# New Perspectives in Therapeutic Vaccines for HPV: A Critical Review

**DOI:** 10.3390/medicina58070860

**Published:** 2022-06-28

**Authors:** Barbara Gardella, Andrea Gritti, Ehsan Soleymaninejadian, Marianna Francesca Pasquali, Gaetano Riemma, Marco La Verde, Maria Teresa Schettino, Nicola Fortunato, Marco Torella, Mattia Dominoni

**Affiliations:** 1Department of Clinical, Surgical, Diagnostic and Paediatric Sciences, University of Pavia, 27100 Pavia, Italy; barbara.gardella@gmail.com (B.G.); mariannepasquali@gmail.com (M.F.P.); matti.domino@gmail.com (M.D.); 2Department of Obstetrics and Gynecology, IRCCS Fundation Policlinico San Matteo, 27100 Pavia, Italy; ehsan.soleymaninejad01@universitadipavia.it; 3Obstetrics and Gynecology Unit, Department of Woman, Child and General and Specialized Surgery, University of Campania “Luigi Vanvitelli”, 81100 Naples, Italy; gaetano.riemma7@gmail.com (G.R.); marco.laverde88@gmail.com (M.L.V.); mariateresa.sche@gmail.com (M.T.S.); nicola.fortunato@libero.it (N.F.); marcotorella@iol.it (M.T.)

**Keywords:** therapeutic vaccine, Human Papillomavirus, CIN

## Abstract

Human Papillomavirus is the main cause of cervical cancer, including squamous cell carcinoma of the oropharynx, anus, rectum, penis, vagina, and vulva. In recent years, considerable effort has been made to control HPV-induced diseases using either prophylactic or therapeutic approaches. A critical review of the literature about the therapeutic Human Papillomavirus vaccine was performed to analyze its efficacy in the treatment of female lower genital tract lesions and its possible perspective application in clinical practice. The most important medical databases were consulted, and all papers published from 2000 until 2021 were considered. We retrieved a group of seven papers, reporting the role of anti HPV therapeutic vaccines against the L2 protein in the order of their efficacy and safety in female lower genital tract disease. In addition, the immune response due to vaccine administration was evaluated. The development of therapeutic vaccines represents an interesting challenge for the treatment of HPV infection of the lower genital tract. Literature data underline that the L2 protein may be an interesting and promising target in the development of therapeutic HPV vaccines, but the possible strengths and the unclear longevity of L2 immune responses are factors to be considered before clinical use.

## 1. Introduction

In the last two decades, the incidence of cervical cancer in Western countries decreased by about one-third. However, this cancer represents the fourth most common cancer in the female population worldwide and about 90% of deaths from cervical cancer occurred in developing countries and the least developing countries largely due to the lack of prevention, diagnosis, and screening [1,2].

The main cause involved in cervical cancer development is Human Papillomavirus (HPV), which represents the most common sexually transmitted infection [3], and more than 200 genotypes were recognized, but alone, HPV16 and HPV18 cause 70% of cervical cancers [4,5,6].

Literature data reported as the annual number of new cases of cervical cancer will increase up to 700,000, and the number of deaths are predicted to reach around 400,000 per year by 2030 [4,5]. For this reason, in 2018, the WHO sustained a new campaign in order to promote the reduction in HPV infection and to permit a decrease in cervical cancer development. The WHO activity aimed to promote the prophylactic vaccination of 90% of young women by 15 years of age, a wide screening program, and the treatment of 90% of women with a diagnosis of Cervical Intraepithelial Neoplasia (CIN) [2].

HPV-associated cancers also include squamous cell carcinoma of the oropharynx, anus, rectum, penis, vagina, and vulva [2,7]. In relation to HPV role in oncogenesis, considerable effort has been made to control HPV-induced diseases using prophylactic or therapeutic approaches [6].

The introduction of the prophylactic anti-HPV vaccine in clinical practice opened a new scenario in the panorama of HPV disease: in 2006, HPV vaccination was approved for the primary prevention of HPV-related diseases in young women, and subsequently, also in women up to 45 years of age [8,9]. Prophylactic vaccines take advantage of the fact that the HPV L1 protein is able to create virus-like particles (VLPs) when expressed in different cell types, which have a strong similarity with native virions [2,10]. These vaccines may prevent HPV infections by eliciting the production of neutralizing antibodies that bind to the viral particles and block their entrance into host cells [2,11]. On the other hand, these vaccines are not effective in the elimination of pre-existing infections, because the target antigens, L1 capsid proteins, are not expressed in infected basal epithelial cells [2,12].

To understand the significance of therapeutic vaccines and their possible role in the management of high-grade intraepithelial lesions or cancer, it is important to know the intrinsic characteristics of the HPV structure and its genome expression. Human Papillomavirus is a non-enveloped icosahedral virus with 72 capsomeres and a diameter of 55 nm. The genome is a circular double strand of DNA with a length of 8000 bp [13].

Analyzing the expression of viral proteins in order of their expression, we can identify three different regions (Figure 1). Nine open reading frames make up the HPV genome, encoding for seven early genes (E1–E7 with regulatory properties in infected cells) and two late genes (L1–L2 encoding capsid proteins). Finally, the viral genome presents long control regions with a cis-acting regulatory sequence involved in the replication and the post-transcriptional controls [13,14]. These three different regions are expressed in different layers of the cervical tissue. The early proteins are expressed in the basal layers. The integration between E1 and E2 and control regions is the first step of viral replication. The early proteins E6 and E7 of the high-risk HPV genotypes are involved in the pathological mechanism of cellular transformation that lead to high-grade cervical intraepithelial lesions and cancer: E6 is able to inactivate tumor suppressor P53, while E7 silences tumor suppressors pRb, p107, and p130. Protein P53 is involved in DNA repair, apoptosis, and growth arrest throughout a cyclin-dependent kinase inhibitor; pRb is implicated in the cell cycle control and modulation, inhibiting the E2F factor [15]. The E7 and pRB and proteins p107 and p130 enable E2F to activate the cell cycle, passing from the G1 phase to the S phase, promoting the transcription of the L1 and L2 proteins [16].

The modification of the cellular control mechanism caused the development of cervical lesions and the concomitant persistence of HPV infection [17]. For this reason, several clinical trials evaluated the possible role of therapeutic anti-HPV vaccines against HPV proteins E6 and E7, the primary oncogenic factors, treating patients with persistent high-risk HPV infection [12] and the highlighted immune response against E6 and E7 proteins, which are used as target antigens for neoplastic progression, but none of these vaccines have been approved by the US Food and Drug Administration or the European Medicines Agency (EMA) so far. Most of the therapeutic vaccines in progress aim to treat pre-existing infections and generally target early proteins expressed in the infected cells and cancer, while limited data are available regarding the other viral proteins [18], for example, the late protein 2 (L2 which) represents an interesting challenge to the development of a proper therapeutic approach. 

The L2 is formed by a sequence of 500 amino acids in length, with a molecular mass of 55 KDa [19]. Each viral capsid contains approximately 72 L2 proteins, with a stable ratio hip with the other late L1 proteins (one L2 protein: five L1 proteins) [20]. The L2 has an important role in modulating mRNA splicing in the epithelial layer, and in addition, it appears to be predominantly hidden below the surface of virions, making the protein configuration unclear [19,21]. Indeed, the L2 is a versatile protein with a complex role in viral HPV infections and host interactions, in particular regarding virion assembly in the early stages of infection and in the delivery of the HPV genome inside the cell nucleus [19,22].

Host proteins promote the cleavage and the phosphorylation of the L2 during virus assembly. This interaction promotes the assistance in capsid conformational change to receptor uptake. For this reason, the L2 is the most important protein to enable efficient virus entry in the host cell with DNA integration through vesicular trafficking and endosomal escape; nuclear transport of the L2 toward the nucleus, viral gene regulation, initial viral gene transcription, morphogenesis and viral capsid assembly [19,21,22].

For this reason, the L2 seems to be a perfect target of therapeutic vaccines, since the ending of several early processes could avoid the persistence and progression of HPV infection. 

Our critical review aims to analyze the possible role and efficacy of a therapeutic vaccine based on late capsid proteins (L2 in particular) in the treatment of HPV, genital lesions and their possible application in clinical practice, especially for the VIN (vulvar intraepithelial neoplasia) and VaIN (vaginal intraepithelial neoplasia), where the treatment is often difficult. Vin, VaIN, and AIN (anal intraepithelial neoplasia) occurred in young patients, where often, the extension of lesion and multifocal localization required excision surgery: for this reason, therapeutic vaccines against L2 capsid proteins could help the clinicians in tailored treatment.

## 2. Materials and Methods

In order to perform a critical review, we consulted several international databases such as PubMed, Cochrane Database of Systematic Reviews, EMBASE, and Web of Science. We researched the following terms and their combinations: “Human Papillomavirus” AND therapeutic vaccines “HPV therapeutic vaccines”, “CIN AND therapeutic vaccines”, “cervical cancer and therapeutic vaccines”, and “late protein HPV therapeutic vaccines”. The research strategy is summarized in Figure 2. We included all available articles published between January 2000 and January 2022 only in English. We performed a systematic search using the Preferred Reporting Items for Systematic Reviews and Meta-Analyses (PRISMA) literature selection method [23]. 

Indeed, two authors independently (A.G. and M.L.V.) analyzed reference lists of recognized papers in order to integrate them into the literature search program. We considered clinical trials regarding therapeutic vaccines based on the L2 protein. Exclusion criteria were the following: single case reports, book chapters, books, conference proceedings, and abstracts. In addition, publications written in a language other than English or in vitro experimental trials were excluded. Finally, we decided to not take into consideration trials regarding anti-HPV vaccines in experimental animal models in the “Results” section. The bibliographic research provided a preliminary set of papers, which were carefully analyzed by two authors (M.D. and M.T.) independently in order to avoid the risk of bias of selection, performance, detection, attrition, and reporting, according to the Cochrane Handbook for Systematic Reviews of Interventions [24,25,26]. For the specific purposes of the present review, we then performed a further selection of the preliminary set of papers, with a more restrictive criterion, i.e., the role of anti-HPV therapeutic vaccines based on the L2 protein in the treatment of cervical lesions and their possible clinical effectiveness and safety. We performed a critical review of this restricted group of articles, which is explained in the last section of the “Results”. The discussion does not intend to be a fully exhaustive dissertation regarding the vast research field of anti-HPV therapeutic vaccines, but it is only a critical review of the main results of the possible role of this category of HPV therapeutic vaccines for the reader’s convenience. 

## 3. Results

We retrieved 58 papers from the preliminary bibliographic search, as we reported in Figure 2. After the elimination of duplicate papers, we found a total of 50 articles (Figure 2). Ten additional articles were found from another source (see flow chart in Figure 2). After the exclusion of 19 manifestly irrelevant records, a careful examination was performed with a selection of a total of 31 papers. In particular, we excluded abstracts, conference proceedings, or previous reviews of this topic. Previous reviews of hormonal treatment were not included. Finally, an additional selection, as previously illustrated, provided a restricted group of seven papers, reporting the role of anti-HPV therapeutic vaccines against the L2; the results of this research are summarized in Table 1, where we report the main significant characteristics of these vaccines. The “Results” section was organized reporting the papers in a chronological sequence in order to discover the steps which led to the development of the clinical trials on therapeutic vaccines against the L2 protein in the gynecological field.

In 2002, de Jong et al. evaluated the safety and immunological response of the HPV-16 L2E6E7 fusion protein (TA-CIN) vaccine, in a double-blind, randomized, and placebo-controlled, dose-escalating phase I trial, which involved 40 healthy volunteers (30 males and 10 non-reproductive females). The population of study was randomized to undergo one of three different doses (25, 125, and 500 micrograms). The immunological profile revealed a significant anti HPV-16 L2E7E6 specific T-cell proliferation induced by TA-CIN in 8/11 patients treated with 533 micrograms of TA-CIN vaccine, which was evaluated with IFN-gamma ELISPOT. In addition, the author reported a significant humoral immunological response with IgG production in 8/32 subjects. All the patients enrolled showed no sign of serious or severe adverse events. Most of the events were reported within 7 days of vaccination. The most frequent adverse effects reported were the following: local injection site reaction and tenderness. Tenderness appeared when higher doses of TA-CIN were administered, while moderate tenderness was higher after vaccination with two high doses. Local reactions (redness and swelling) were a rare adverse effect. The most frequently systemic adverse moderate events were headache and fatigue, without a correlation with the dose of TA-CIN or the number of administrations [27].

In 2003, Davidson et al. in a Phase II trial evaluated the immunological response and clinical features of patients with HPV-16 related high-grade Vulva Intraepithelial Neoplasia (HG VIN) treated previously with TA-HPV, which was based on modified HPV-16 and 18 E6–E7 proteins. In this trial, 18 women were enrolled and 10 demonstrated an increase in HPV-16 specific T-cell response. In one patient, there was a complete regression of vulvar lesions with histological confirmation and viral clearance by PCR evaluation. Eight patients reported a reduction in vulvar lesions of at least 50% of diameter, and in six of these women, the authors demonstrated a reduction in viral load; four patients showed a decrease in symptoms. Vaccine immunogenicity was confirmed by the increase in serological and cell-mediated immunity after vaccination. All 18 women reported vaccine-specific IgG, and in 17/18 patients, there was a >10-fold increase from the pre-vaccination titer. All patients showed an IFN-gamma ELISPOT response. Seven women reported vaccine-specific T cells after vaccination. In 14 of these women, an increase from the pre-vaccination level of T cells was reported. Regarding HPV-specific immunity, no responses were reported in any patient to the E6. One patient showed an increased E7 T cell response but with low level T cell responses to three different HPV-16 E7 peptides, three patients reported an increase in response but with low level T cell responses to two different HPV-16 E7 peptides, and three patients showed the same increased response but with low level T cell responses to one of the HPV-16 E7 peptides after vaccination. Serological responses were evaluated to HPV-16 and 18 E6 and E7 by ELISA. Two patients reported a clearly increased HPV-specific serological response detected at 4 weeks after vaccination, but IgG levels returned to the pre-vaccination level by 12 weeks post-vaccination. Clinical responding women had significantly more lesion-associated CD1a, CD4, and CD8-positive immune cells before vaccination than non-responders. The authors underlined that the therapeutic vaccine response may be influenced by the local immune response: indeed, responder patients reported a high level of CD4+ and CD8+ associated with lesions. Viral load was evaluated by real-time PCR. ELISPOT using HLA-A2 binding peptides was performed in order to investigate HPV-specific immunity. T cell proliferation was investigated with an HPV-16 L2E6E7 fusion protein. Antibodies were measured by ELISA. Lesion-infiltrating CD4+ and CD8+ immune cells response was evaluated by immunohistochemistry [15]. 

In 2004, Smith et al. evaluated the immunogenicity of heterologous vaccination in 29 women with high-grade ano-genital intraepithelial neoplasia: each patient underwent three prime doses of TA-CIN (533 μg) i.m. at four weekly intervals, which was followed after 4 weeks with a single dose of TA-HPV by dermal scarification. In 17 patients, a TA-CIN-specific response after prime or prime-boost vaccination was reported. Ten of these 17 patients reported a definite vaccine induced response. Regarding the proliferative responses against the oncoproteins E6 and E7 using recombinant HPV-16 GST-E6 or GST-E7 proteins, there was only a significant increase in the response for GST-E6 at week 16. IFN-γ ELISPOT assays highlighted HPV-16 E6-specific T-cell responses in nine of 25 patients, which was induced by vaccination. In two women, HPV-16 E7-specific T cells were enhanced after all vaccinations. Three patients reported enhanced HPV-18 E6-specific T-cell activity post-vaccination at week 20. No significant changes were reported in serological responses to HPV-18 E6 or E7 after vaccination. The responses to HPV-16 E6 or E7 appeared transient: the level peaked at week 16, and it declined by weeks 20 and 24, which was in accordance with the single booster vaccination with E6 and E7 of TA-HPV at week 12. Indeed, the authors reported a significant response in the study group in both humoral and cellular immunological response, respectively; nevertheless, the study did not show a correlation between systemic HPV-16-related immunity and clinical outcomes. In fact, among 19 patients with stable disease, eight reported immunogenic responses, and in the four patients with disease progression, two did not show a T cell response, and two revealed an absence of both T cell and humoral responses [28].

In the same year, Davidson et al. enrolled in a TA-CIN extension trial ten women affected by HPV-16-related VIN. These patients were treated primary with heterologous vaccination: prime boosters of TA-HPV (virus encoding HPV-16/18 and E7 vaccine) and subsequently with TA-CIN (HPV-16 L2E6E7 fusion protein vaccine). Three patients reported a clinical response, while six patients maintained stable disease. Two patients reported a partial or complete clinical response, which was confirmed by the reduction in lesion diameter of ≥50% following vaccination with TA-CIN. Six patients maintained stable disease, one patient reported significant symptomatic relief following vaccination, while one patent reported disease progression (increase in lesion diameter of ≥25% post-vaccination). Regarding serological responses, IgG levels did not demonstrate a significant change during the study. Nearly all (9/10) patients reported at least a two-fold increase in HPV-16 and/or 18 E6- and/or E7-specific IgG after vaccination with TA-CIN. In eight patients, an increased HPV-16 E7-specific IgG response was reported; in six women, an increased HPV-16 E6-specific IgG response was reported, and in one patient, an increased HPV-18 E6-specific IgG response after vaccination was not demonstrated. Regarding HPV-specific proliferative responses, all patients demonstrated a proliferative response. Analyzing HPV-specific ELISPOT responses, three patients reported a response to one or more HPV-16 E7 peptide before and after vaccination. Two patients reported an increase in HPV-specific peptide responses following vaccination. The results reported as the combination of TA-HPV and TA-CIN as heterologous vaccination may be an interesting combination in HPV-induced lesion treatment due to the immunogenicity conferred by the vaccine [29].

In 2006, Fiander et al. reported in a multi-center Phase II trial the effectiveness of a prime boost regimen for the treatment of HPV-related diseases. In prime-boost vaccination, the authors evaluated heterologous HPV vaccines in the management of ano-genital intraepithelial neoplasia. In this study, patients were vaccinated with three doses of TA-CIN followed by one dose of a recombinant vaccine with virus-encoding HPV-16 and 18 E6/E7 (TA-HPV). Twenty-nine women were recruited for the study: 2 with VAIN 3, 27 with VIN3, and 2 patients with VIN3 also had anal intraepithelial neoplasia. A complete response was seen in 1 patient, partial response in 5, stable disease in 18, and progression in 5 of them. None of the patients developed invasive lesions [30].

In 2010, Dayana et al. combined in a phase II clinical trial the TA CIN vaccine and Imiquimod (topical immunomodulatory) in 19 women to treat VIN2 and 3. The authors reported a complete regression of VIN in 6/19 subjects (32%) at week 10, in 11/19 patients 58% at week 20, and in 12/19 at week 52 (63%). At week 20 in responder patients, there was a significant increase in the response of CD4+ and CD8+ (*p* = 0.03 and *p* = 0.03). The authors concluded that the modulation of the local and systemic immune response plays a role in the therapeutic vaccine effect [31].

In 2015, a clinical trial regarding tissue antigen-CIN (TA-CIN) was approved, and it is currently ongoing. The trial is a randomized, multi-center, open-label pilot study. The first aim of this trial is to provide evidence for the safety and feasibility of the vaccine and the different immune responses when the dose is administered at different locations such as the thigh or arm in order to determine the site that can elicit the most potent immunological effect. Furthermore, the study aims to evaluate the level of circulating antibodies to HPV16 E6, E7, and L2, as well as the level of circulating T cells and the mononucleocyte response in the peripheral blood before and after vaccination (time frame: up to 4 years). In this pilot study, a single dose level (100 µg) is administered in order to evaluate the assessment for the safety and tolerability of administering the TA-CIN vaccine three times to either the arm versus the thigh of patients who have previously been treated for HPV-16-related cervical cancer in the past year with no evidence of disease recurrence. A total of 14 patients have been enrolled to demonstrate the safety of the TA-CIN vaccine as adjuvant therapy. The study design aims to evaluate the minimal or non-systemic toxicity of the administration of the vaccine. TA-CIN will be given as a single intramuscular injection every 4 weeks for a maximum of three times. The location of the injection (arm or thigh) is casual. Patients will be evaluated for safety and response to treatment during this period. The follow-up period includes the following: clinical visits are performed at different times after the administration to evaluate the clinical response [32].

## 4. Discussion

### 4.1. HPV Vaccines Based on L2 Protein: Rationale

The development of therapeutic vaccines represents an interesting challenge for the treatment of HPV infection of the lower genital tract. Indeed, the development of therapeutic vaccines may constitute a valid clinical treatment option; nevertheless, there are no vaccines of this nature currently available in clinical practice [13,14].

Experimental animal models demonstrated that neutralizing antibodies represent the main goal of immunological protection because they were found in the microenvironment of microlesions in the cervical epithelium, which are the “door” of viral entrance to infect the basal lamina of cervical tissue [12]. Currently, the prophylactic vaccines against L1 did not show efficacy against pre-existing HPV lesions because the L1 was not expressed in the basal layer. On the other hand, therapeutic vaccines against only E6 and E7 failed for long-term immunological responses. For this reason, pre-clinical research aimed to find other viral proteins more suitable to elicit immunological response, such as the L2 capsid protein. 

Vaccines against the minor structural protein L2 demonstrated an improvement of efficacy because they conserved N-terminal epitopes, which may promote the increase in the level of cross-type neutralizing responses and protection against the skin and genital HPV infection, especially against the first 130 amino acids of the L2 peptide [16]. 

As previously reported, the goal of the development of the next generation of HPV vaccines is to block the transcription and integration of viral genome into host cells. Since the L2 protein is not able to form VLPs as L1 proteins, the fusion of the L2 with early viral proteins (particularly, E6 and E7) may promote the elicitation of a therapeutic immunological response [33]. In order to enhance the immunogenicity, experimental projects aim to combine the L2 peptide conjugated with thioredoxin or concatemers of the L2 protein fused to a self-adjuvant protein (flagellin) [34,35,36,37,38,39]. More recently, new experimental projects tested an L2 polypeptide and a heptamerizing coiled-coil polypeptide OVX313 linked to the nanoparticle thioredoxin-L2-OVX313 [40,41,42]. These approaches have demonstrated the immunogenicity of L2 peptides, and more specifically, they have underlined the possible role of this protein in the next generation of vaccines [12].

### 4.2. HPV Immunological Response and L2 Therapeutic Vaccines

The progression of HPV anogenital lesions and the development of high-grade pre-neoplastic lesions mirror a dysregulation of the immune system associated with an increase in E6 and E7 expression, with a consequent augmentation of HPV gene expression and loss of cell differentiation of the basal layer [43]. During HPV infection, the immunological response is based on humoral and cell-mediated activation; for this reason, a large part of the population infected by HPV can delete the viral infection and favor the clearance of HPV. Experimental data reported that the protein E2 can be found occasionally in blood mononuclear cells in healthy subjects, but even in patients with genital HPV lesions and the immunological response against HPV-16, E2 was the same in healthy and in patients with cervical disease [44,45].

After HPV tissue attack, host cells activate the innate and adaptive immune systems in order to eliminate the infection. Macrophages, Langerhans cells, and the natural killer cells try to inhibit HPV proliferation and integration with Toll-like receptors (TLRs) expression. TLRs are able to recognize the viral components and promote the transcription of factor-like nuclear factor kappa B (NF-κB) and interferon response factor-3 (IRF3) to increase the proliferation of pro-inflammatory cytokines and antiviral cytokines (interferon-gamma tumor necrosis alpha and interleukin 2) [46]. 

Literature data supported the opinion that systemic T cell response against HPV infection can explain the different behavior of HPV lesions. Major histocompatibility complexes (MHC I–II) represent the basis of the acquired immunological system. MHC I exposes the antigen to cytotoxic T cells CD8+, while MHC II exposes the antigen to helper T cells CD4+. 

In particular, Th1 CD4+ T-cells are specific for HPV-16 E6, E7, and E2, because they are found in the peripheral blood of immunocompetent individuals with HPV confirmed infection [47]. Interestingly, a HPV-18 specific T cell response was detected only in healthy patients and not in subjects with confirmed infection sustained by this HPV genotype; this is probably the mechanism that determines viral persistence. In addition, higher levels of intraepithelial CD8+ T Cells in cancer cervical tissue promote a decrease in lymph node metastasis in patients with early-stage cervical cancer [48,49].

Finally, literature data supported the hypothesis that the natural anti-HPV cellular response causes mild protection against HPV genotypes involved in lower genital tract infections, and the natural immunological response appears lower than the protection given by vaccine administration [50].

Therapeutic vaccine technology is based on two fundamental points: the induction in the host of a T cell response against E6, E7 and L2 fusion proteins in order to improve effectiveness and to promote the lesion regression and the ability of an immunological response to identify the site of HPV infection [51].

As we reported, the TA-CIN vaccine is able to promote an HPV-16 specific immunological response and high levels of anti HPV-16-neutralizing antibodies. Indeed, Gambhira et al., using the serum derived from previous clinical trials of Phase I and II, evaluated the power of the HPV E6E7L2 vaccine to induce the L2 specific response. The authors found a significant seroconversion and higher specific responses for HPV-16 and HPV-18 neutralizing antibodies, increasing vaccine dose in healthy volunteers. Therefore, the efficacy of the TA-CIN vaccine showed a reduction in high grade ano-genital intraepithelial lesions rather than in healthy volunteers [52].

During the Phase I/II studies, a regression of the lesions and an important resolution of symptoms were found, and a strong response from CD4 + and CD8+ T cell produces the clearance of the virus, while in non-responders, there is a prevalence of T-Reg cells [49,53,54,55,56,57,58,59,60,61,62]. 

Therapeutic vaccines based on the fusion of viral protein technology are not only direct against high-grade cervical lesions or cervical cancer but also against genital warts. In this setting, clinical trials evaluated the tissue antigen–genital warts vaccine (TA-GW) based on a fusion protein of HPV-6 E7 and L2, and it was able to stimulate cellular and humoral immunity in order to promote the regression of existing genital warts and prevent their recurrence [63,64]. Two previous studies investigated the immunological response of TA-GW combined with Alhydrogel^®^ to induce a T cell and a humoral response against L2 and E6. The limits of these data are the frequent spontaneous rate of regression of genital warts [65,66], but this research opens the way to use the treatment in laryngeal juvenile respiratory papillomatosis of children and in vulvar, vaginal and anal HPV lesions, where the treatment is often repetitive and surgical.

From a perspective point of view, it is interesting to analyze the feasible application of DNA vaccination conjugated with the current vaccine technology. Peng et al. in experimental animal models tested a DNA vaccine encoding for calreticulin in addition to E6, E7 and L2 of HPV-16 in order to study the efficacy in immunocompromised subjects with low CD4 T-cell levels. The safety and efficacy of the DNA vaccine may overcome the risk of administration of life vectors in immunodeficient subjects inducing an HPV cytotoxic T cell response. Interestingly, the T cell response promotes subcutaneous tumor regression and removes circulating tumor cells, and for this reason, DNA vaccines appear to offer several advantages [67,68,69,70]. 

## 5. Conclusions

Finally, the L2 protein may be an interesting promising target in the development of a broad-spectrum HPV vaccine, but the possible strengths and the unclear longevity of the L2 immune responses are factors to be considered in clinical use. 

## Figures and Tables

**Figure 1 medicina-58-00860-f001:**
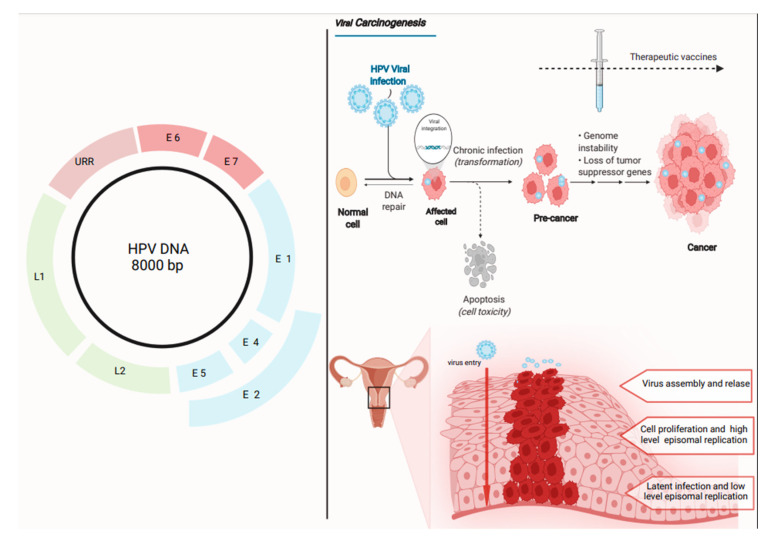
HPV genome structure (**left**) and the HPV pathway of infection and neoplastic transformation of cervical epithelial cells with the possible application of therapeutic vaccines (**right**). Legend: URR: Upstream Regulatory Region; E: early protein; L: late protein. Figure created with BioRender.com (accessed on 15 February 2022).

**Figure 2 medicina-58-00860-f002:**
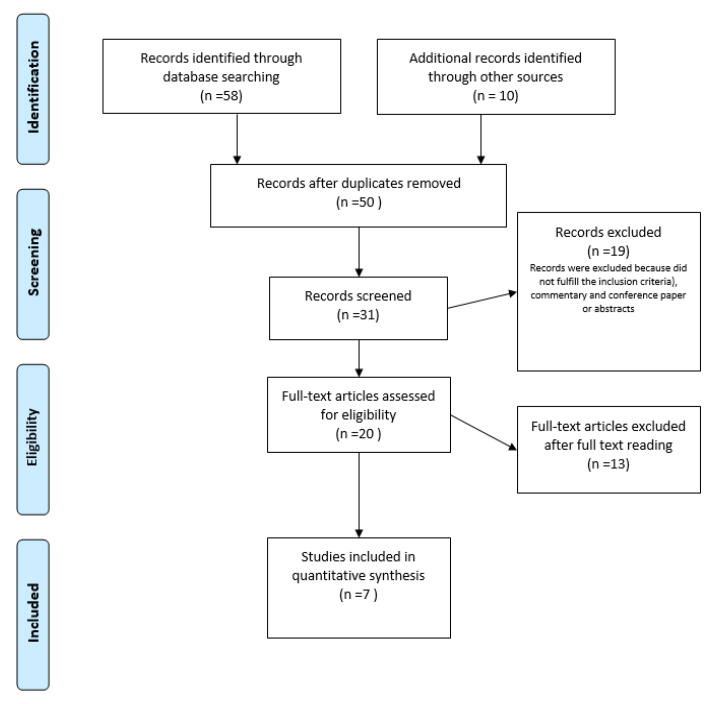
PRISMA flow diagram of the study selection process.

**Table 1 medicina-58-00860-t001:** Experimental trials about anti HPV therapeutic vaccines based on L2 proteins.

Study Author and Years	Composition	Main Advantages Reported	Develop Phase
De Jong et al., 2002 [27]	TA-CIN vaccine (recombinant HPV-16 L2 E6 E7)	Immunological profile: revealed a significant anti HPV-16 E6 and E7 T cell response in 8/11 patients treated with TA-CIN vaccine. Immunological response with IgG production was reported. No reported serious or severe adverse events.	Phase I
Davidson et al., 2003 [15]	Therapeutic vaccine based on modified HPV-16 and 18 E6–E7 proteins	Evaluation of vaccines in vulvar intraepithelial neoplasia. Ten patients demonstrated an increase in HPV-16 specific T-CELL response. In one patient, there was a complete regression of vulvar lesions and viral clearance by PCR evaluation. In eight patients, a reduction in vulvar lesions of at least 50% of diameter was reported, and in 6 of these women, a reduction in viral load. Four patients showed a decrease in symptoms. All patients reported an increase in both serological and cell-mediated immunity.	Phase II
Smith et al., 2004 [28]	TA-CIN vaccine (recombinant HPV-16 L2E6E7)	Evaluation of vaccine application in high-grade ano-genital intraepithelial neoplasm. Regarding the proliferative responses against the oncoproteins E6 and E7, there was only a significant increase in the response for E6. HPV-16 E6-specific T cell responses in 9 patients, induced by vaccination. In two women, HPV-16 E7-specific T cells were enhanced after all vaccinations. Three patients reported enhanced HPV-18 E6-specific T-cell activity after vaccination. No significant changes were reported in serological responses to HPV-18 E6 or E7 after vaccination. Among 19 patients with stable disease, 8 reported an immunogenic response, and in the 4 patients with disease progression, two did not show a T-cell response and two revealed an absence of both a T-cell and a humoral response	Phase II
Davidson et al., 2004 [29]	TA-CIN vaccine (recombinant HPV-16 L2 E6 E7) and TA-HPV (vaccine based on virus encoding HPV16/18 and E7)	Ten women affected by HPV-16 related VIN were enrolled. Three patients reported a clinical response, while 6 patients maintained stable disease. Two patients reported a partial or complete clinical response, which was confirmed by the reduction in lesion diameter of ≥50%. Six patients maintained stable disease, one patient reported significant symptomatic relief following vaccination, while one patient reported disease progression (increase in lesion diameter of ≥25% post-vaccination). IgG levels did not demonstrate a significant change. Regarding HPV-specific proliferative responses, all patients demonstrated a proliferative response.	Phase II
Fainder et al., 2006 [30]	TA-CIN vaccine (recombinant HPV-16 L2 E6 E7) and TA-HPV (vaccine based on virus encoding HPV16/18 and E7)	Patients with ano-genital intraepithelial neoplasia were vaccinated with 3 doses of TA-CIN followed by one dose of a recombinant vaccine with virus encoding HPV-16 and 18 E6/E7 (TA-HPV). Two with VAIN 3, 27 with VIN3 and 2 patients with VIN3 also had anal intraepithelial neoplasia. A complete response was seen in 1 patient, partial response in 5, stable disease in 18, and progression in 5 of them. None of the patients developed invasive lesions.	Phase II
Dayana et al. 2010 [31]	TA-CIN vaccine (recombinant HPV-16 L2 E6 E7) and Imiquimod	Women with VIN2 and VIN3 were enrolled. A complete regression of VIN in 12/19 at week 52 of treatment (63%). In responder patients, there was a significant increase in a CD4+ and a CD8+ response. The authors concluded that the modulation of local and systemic immune response plays a role in the therapeutic vaccine effect.	Phase II
Safety and Feasibility of TA-CIN Vaccine in HPV16 Associated Cervical Cancer [31]	TA-CIN vaccine (recombinant HPV-16 L2 E6 E7)	(ongoing study) The first aim of this trial is to provide evidence for the safety and feasibility of the vaccine and the different immune responses when the dose is administered at different locations such as the thigh or the arm in order to determine the site that can elicit the most potent immunological effect. Secondary aims were the evaluation of the level of circulating antibodies to HPV-16 E6, E7, and L2, the level of circulating T-Cells and the mononucleocyte response in the peripheral blood before and after vaccination	Phase I

## Data Availability

Not applicable.

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
