# Peer review of "New Perspectives in Therapeutic Vaccines for HPV: A Critical Review"

_medicina, 2022, doi:10.3390/medicina58070860_

Round 1

Reviewer 1 Report

Reviewer comments for medicina-1706769
The authors have synthesized a systematic review from available literature on the potential of L2 protein of the Human Papiloma Virus as a viable therapeutic vaccine candidate. While HPV is a leading cause of cervical cancer in women globally, the availability of a vaccine doesn’t treat all the serotypes/variants of the virus, nor does it treat preexisting infection. On this ground, the case for a prophylactic vaccine is strong. However, the authors have written this in a very incoherent manner with several shortcomings that cannot be rectified just by extensive edits. It is really not my job as a reviewer to improve the English in the current manuscript. However, I have provided comments below for the authors to work on.
General comments –
1) The problems begin with the language used. The English used us subpar and doesn’t adhere to any manuscript standard whatsoever. It looks like no one bothered to proofread this manuscript before submitting. Moreover, I have a very strong suspicion that the document was originally written in a different language and then translated using some app or program. While this in theory translated the text, the nuanced meanings of words and sentence construction completely went for a toss.
2) Authors need to consult an oncologist and a virologist both on how to frame their text specifically regarding virus structure. The manuscript has a high number of conceptual errors.
3) The review has been written in a very incoherent manner where several sentences in the Discussion should have been included in the introduction without which, I as a reader or reviewer, am left wondering why did the authors go through so much trouble and not explain the virology in the introduction?
4) Entire sentences on lines 63, 148, 267, 300, 335, 339, 358 make no sense whatsoever.
Line by line comments –
Authors don’t have to write background, method, conclusions in the abstract.
Line 26 - Is it a tumor or a legion? Tumor could also be benign.
Line 37 - favorite? sustained? This is what I mean by English being subpar. Those words make no sense in that sentence.
Line 39 and 47 – Authors alternate between the terms girls, women, females, subjects and patients – use uniform nomenclature.
Line 58, 59 - What is a high-grade genotype? Is the US FDA the global standard for vaccines? Does Italy or the EU not have its own FDA equivalent?
Line 70 - This is a good point and authors should bring it out more. The treatment for these cancerous conditions is actually very destructive and hence need for therapeutic vaccines.
Line 82 - Authors need to elaborate more on what is the L2 protein in the virus and host concept. I would recommend consulting with a virologist. This point will come up again in the Discussion.
In general 7 manuscripts for a systematic review is rather low.
Line 126 – What was the outcome of this phase I trial?
Line 134 - Why were adverse events not reported if the phase trial was about "safety"? What is E cell in immunological terms? What do authors mean by significant humoral response? How was it quantified in the 2002 study?
Line 150 - So there was a significant response - how was it quantified? And if it was significant, why was there no correlation between immunological response and clinical outcomes?
Line 152 - What do the authors mean by "reported a response"? Did the trials count their antibodies? B and T cell numbers? Any other cytokines or interferon counts?
Line 157 - How are readers supposed to know what a heterologous prime boost is?
Line 161 – “The authors concluded that …” - Why and what does this mean?
Table 1 - What new information does this table provide that the authors have not included in the text above? The Van der Burg et al. trial - The antigen response was not correlating to clinical outcomes in several of these studies. That means the vaccine is illiciting a response but not the correct one?
Discussion –
Line 185 – I am not convinced by this argument. Why are therapeutic vaccines necessary if regular vaccines work well?
Line 188 - In structural biology, structure implies function and more so for viruses. Authors should have 1) checked the validity of the sources where they mention structural information and 2) included this in the introduction. I am not an expert in HPV and I have to wait till the discussion to know what part of the virus is used in vaccine design. This is very poor writing. Also, reference 25 is not what the authors should be citing for virus structure. Please find an electron microscopy paper where a structure of any HPV variant or serotype, either as full virus or as a capsid has been solved and cite that.
Line 193 – Authors should seriously consult with a virologist before they write things like this. If the virus genome has 9 ORFs that means they all can be translated and transcribed in parallel at the same time. So, the "order of expression" argument is invalid.
Line 195 - What is regulatory property in infected cell? What is post transcriptional control?
Line 213 - Firstly, there was no need to go into gory details of virus structure in a paper that is a critical review about prophylactic vaccines. Then authors have written it incorrectly. Its 360 copies spread in 72 capsomers, so 5 L1 copies per pentameric capsomer. Structure of L2 was solved very recently using cryoem in 2021 - https://www.nature.com/articles/s41598-021-83076-5#Abs1
Line 232 and discussion in general - Half the discussion is information that should be included in the introduction. This is actually a well written paragraph but completely out of place. Also, are authors talking about therapeutic vaccines or prophylactic vaccines here?
Line 247 - So authors want to say that L2 fused with other proteins is a better candidate than L1? Or that both L1 and L2 based vaccines should be tested in trials? What is the point authors are trying to make here?
Line 261 - The results described in the manuscript and the discussion are largely not correlated. It seems like 2 disjointed manuscripts were hastily appended together.
Line 293 - In a very long winded, indirect sort of way, are the authors saying that the cellular response is the main protection against HPV infection because the humoral response is slow? Is that what the authors mean? And that the vaccine is basically expediting the humoral response while also helping the cellular response along? Why is it so difficult to make what the authors have written?
Line 304 - Why are authors mentioning p values here? They don’t tell me anything about the actual immune response quantification other than their statistical error being insignificant.
Line 320 - I am still not sure how authors are so sure that antibodies are illicited against or by presence of L2. How was this tested?

Author Response

1) The problems begin with the language used. The English used us subpar and doesn’t adhere to any manuscript standard whatsoever. It looks like no one bothered to proofread this manuscript before submitting. Moreover, I have a very strong suspicion that the document was originally written in a different language and then translated using some app or program. While this in theory translated the text, the nuanced meanings of words and sentence construction completely went for a toss.

English editing was performed by a  USA native speaker

2) Authors need to consult an oncologist and a virologist both on how to frame their text specifically regarding virus structure. The manuscript has a high number of conceptual errors.

As the reviewer stated, he’s not an expert in HPV. In fact he doesn’t know HPV disease is studied and managed by gynecologists. For this reason authors think oncological or virological consultation is unnecessary for this paper

3) The review has been written in a very incoherent manner where several sentences in the Discussion should have been included in the introduction without which, I as a reader or reviewer, am left wondering why did the authors go through so much trouble and not explain the virology in the introduction?

The virological characteristics of HPV were reported in the introduction. 

4) Entire sentences on lines 63, 148, 267, 300, 335, 339, 358 make no sense whatsoever. 

English editing was performed

5) Authors don’t have to write background, method, conclusions in the abstract.

Abstract was corrected according  to the reviewer's suggestion. 

 6) Line 26 - Is it a tumor or a lesion? Tumor could also be benign.

 Lesion term represents, in this paper, the cervical dysplastic modification HPV- related according to LAST classification (Obstet Gynecol Clin North Am. 2013; 40(2): 225–233. doi: 10.1016/j.ogc.2013.02.008. The LAST Project and its Implications for Clinical Care. Nuño  T, and García F). 

The aim of the therapeutic vaccine is to prevent the progression from HSIL (High grade Squamous Intraepithelial Lesion) to cervical cancer.

7) Line 37 - favorite? sustained? This is what I mean by English being subpar. Those words make no sense in that sentence.

A correction was made

8) Line 39 and 47 – Authors alternate between the terms girls, women, females, subjects and patients – use uniform nomenclature.

The authors uniformed the nomenclature, nevertheless  the use of varied and not repeated terminology is not a conceptual error.  

9) Line 58, 59 - What is a high-grade genotype?

The author added  “Risk” in order to correct the purpose.

10)  Is the US FDA the global standard for vaccines? Does Italy or the EU not have its own FDA equivalent?

A correction was made.

11) Line 70 - This is a good point and authors should bring it out more. The treatment for these cancerous conditions is actually very destructive and hence need for therapeutic vaccines.

The introduction of therapeutic vaccines is needed to avoid the development of preneoplastic lesions with all the implications. The treatment is only a part of the problem of HPV disease, the relapse of lesion, the persistence of HPV infection promotes the oncogenesis and the social cost of screening and treatment.

12) Line 82 - Authors need to elaborate more on what is the L2 protein in the virus and host concept. I would recommend consulting with a virologist. This point will come up again in the Discussion.

The description of L2 was reported in the introduction

13) In general 7 manuscripts for a systematic review is rather low.

In literature there are only 7 manuscripts in humans about therapeutic HPV vaccine against L2 protein because it’s very a new field in expansion.

14) Line 126 – What was the outcome of this phase I trial?

an addition regarding this trial and the correct reference were reported, also in table 1. 

15) Line 134 - Why were adverse events not reported if the phase trial was about "safety"? What is E cell in immunological terms? What do authors mean by significant humoral response? How was it quantified in the 2002 study?

Adverse events were reported in the manuscript.  E cell was a mistake. It was corrected. Humoral response represents the antibody-mediated immunity and the quantification method was reported.

16) Line 150 - So there was a significant response - how was it quantified? And if it was significant, why was there no correlation between immunological response and clinical outcomes?

An addition regarding the methods to investigate the response was added. In addition a clinical response there was such as an immunological response.  

17) Line 152 - What do the authors mean by "reported a response"? Did the trials count their antibodies? B and T cell numbers? Any other cytokines or interferon counts?

 The authors increase the description of the paper with more details regarding the response to TA HPV vaccine.  

18) Line 157 - How are readers supposed to know what a heterologous prime boost is?

A correction was made

19) Line 161 – “The authors concluded that …” - Why and what does this mean?

 A correction was made.

20) Table 1 - What new information does this table provide that the authors have not included in the text above? The Van der Burg et al. trial - The antigen response was not correlating to clinical outcomes in several of these studies. That means the vaccine is eliciting a response but not the correct one?

Table 1 was revised and enriched. In the author opinion the table may help the reader to analyze the most significant results of the paper

21) Line 185 – I am not convinced by this argument. Why are therapeutic vaccines necessary if regular vaccines work well?

 The “ regular vaccines” works well in naïve patients, but in sexually active people works in only 40-60% of cases (N Engl J Med. 2007 May 10;356(19):1915-27. doi: 10.1056/NEJMoa061741. Quadrivalent vaccine against human papillomavirus to prevent high-grade cervical lesions FUTURE II Study Group). The aim of prophylactic vaccine is to prevent the infection, while the aim of therapeutic vaccine is to prevent the progression from infection to cancer in HPV infected people.

22) Line 188 - In structural biology, structure implies function and more so for viruses. Authors should have 1) checked the validity of the sources where they mention structural information and 2) included this in the introduction. I am not an expert in HPV and I have to wait till the discussion to know what part of the virus is used in vaccine design. This is very poor writing. Also, reference 25 is not what the authors should be citing for virus structure. Please find an electron microscopy paper where a structure of any HPV variant or serotype, either as full virus or as a capsid has been solved and cite that.

I agree with referee  about the need of underline the part of the virus used in vaccine design in the introduction, as we reported.  A reference, as requested was added and corrected. 

23) Line 193 – Authors should seriously consult with a virologist before they write things like this. If the virus genome has 9 ORFs that means they all can be translated and transcribed in parallel at the same time. So, the "order of expression" argument is invalid,.

HPV genome encodes 8 major proteins: 6 located in Early regions (E), and 2 in late regions (L). This  classification does not represent the order of protein expression. 

24) Line 195 - What is regulatory property in infected cell? What is post transcriptional control?

The early proteins are regulatory in function: for example they play roles in HPV genome replication and transcription, cell cycle, cell signaling,apoptosis control, immune modulation, and structural modification of the infected cell.  The post-transcriptional control represents a set of biological processes of regulation at the level of RNA processing, nuclear export, mRNA stability and translation of viral RNAs. An array of gene promoters in the HPV early region may be actively engaged in transcription of the genome and controlled in response to epithelial differentiation

25) Line 213 - Firstly, there was no need to go into gory details of virus structure in a paper that is a critical review about prophylactic vaccines. Then authors have written it incorrectly. Its 360 copies spread in 72 capsomers, so 5 L1 copies per pentameric capsomer. Structure of L2 was solved very recently using cryoem in 2021 - https://www.nature.com/articles/s41598-021-83076-5#Abs1

the correction was reported , in accordance with the reviewer's suggestion. 

26) Line 232 and discussion in general - Half the discussion is information that should be included in the introduction. This is actually a well written paragraph but completely out of place. Also, are authors talking about therapeutic vaccines or prophylactic vaccines here?

A part of discussion was added in the introduction. in the discussion we discussed about  prophylactic vaccines in order to underline the difference with  therapeutic vaccines.

27) Line 247 - So authors want to say that L2 fused with other proteins is a better candidate than L1? Or that both L1 and L2 based vaccines should be tested in trials? What is the point authors are trying to make here? 

Vaccine against L1 protein are prophylactic, while Vaccine against L2 protein could work well in ongoing HPV infection to prevent the persistence and the progression of intraepithelial lesion. We agree with referee that L1 strategy is very simple to avoid the infection,but inefficacy against the previous infection. The authors sustained the hypothesis that a therapeutic vaccine strategy based on L2  may be possible, as the data of literature reported suggested.  

28) Line 261 - The results described in the manuscript and the discussion are largely not correlated. It seems like 2 disjointed manuscripts were hastily appended together.

We reviewed the entire manuscript in order to merge results with the discussion.

29) Line 293 - In a very long winded, indirect sort of way, are the authors saying that the cellular response is the main protection against HPV infection because the humoral response is slow? Is that what the authors mean? And that the vaccine is basically expediting the humoral response while also helping the cellular response along? Why is it so difficult to make what the authors have written?

Recently the authors (Vaccine, 2022 )issued a review on the recent progression in cellular and humoral immunity studies during the progression of HPV-related cancers. In HPV infection many factors play a key role in primary contact with virus, in the replication and finally in the integration in the host cells to promote oncogenesis.Again, these L1-vaccines are not effective in the elimination of  pre-existing infections, because the target antigens, L1 capsid proteins, are not expressed in infected basal epithelial cells [2, 12]. 

The therapeutic vaccines may favor  the immunological humoral response against HPV infection,  as the data of clinical data reported.

50) Line 304 - Why are authors mentioning p values here? They don’t tell me anything about the actual immune response quantification other than their statistical error being insignificant.

A correction was made.

31) Line 320 - I am still not sure how authors are so sure that antibodies are elicited against or by presence of L2. How was this tested?

a correction was made and the purpose was corrected 

Reviewer 2 Report

The authors of the present review article suggest L2 proteins for vaccine development against HPV. The article is focused avoiding general ideas which is good for a review article. The sequence of the methodology followed, in terms of studies included, is reasonably explained. There a re only a few points that follow that should be modified for better presentation of the main ideas

The abstract is too short. The conclusion is provided without any previous information explain how it was derived. I suggest to add a few things regarding how the conclusion and final suggestion were obtained. 

The scope of this review paper in the introduction is well described. However, nothing is said in the introduction regarding the L2 before the scope of the study. The authors should explain why they have chosen this vaccine to analyse as well as a few things regarding this L2

In lines 90-91, why was this decision adopted? Why did the authors decide not to include experimental animal models? Please provide an explanation

In lines 190-198 I suggest the authors to illustrate the HPV genome in a Figure

Line 359: “will a clinical trials which can demonstrate” something is missing, please rephrase

Author Response

1) The abstract is too short. The conclusion is provided without any previous information explain how it was derived. I suggest to add a few things regarding how the conclusion and final suggestion were obtained. 

The correction was made and the abstract was enriched

2) The scope of this review paper in the introduction is well described. However, nothing is said in the introduction regarding the L2 before the scope of the study. The authors should explain why they have chosen this vaccine to analyze as well as a few things regarding this L2

In the introduction was described L2 characteristics and significance

3) In lines 90-91, why was this decision adopted? Why did the authors decide not to include experimental animal models? Please provide an explanation

The authors in the materials and methods considered an exclusion criteria the animal and in vitro experimental data. This consideration was performed in order to underline the few data about therapeutic vaccines based on L2 nowadays available. In the other hand this decision may permit to the reader to focus on the results in human subjects and the importance of these data in this field.  

4) In lines 190-198 I suggest the authors to illustrate the HPV genome in a Figure

A figure was created.

5) Line 359: “will a clinical trials which can demonstrate” something is missing, please rephrase

A correction of this purpose was made

Round 2

Reviewer 1 Report

General comments –
English continues to be a problem. I recommend an extensive round of proof reading. I don’t understand how authors are making such disjointed paragraph and often single lines are made into paragraphs. Absence of line numbers makes giving comments and assessing the modified manuscript very difficult. I still find weak and incoherent arguments in favor of using L2 as a vaccinate component. I don’t doubt the potential of HPV L2 protein as a good immunogen, I just disagree with the way authors have made a case for it.

Line by line comments -
Introduction – “For this motivation…” what do authors mean by motivation? Also, this highlighted portion still doesn’t tell me why therapeutic vaccines are the need of the hour.
Page 3 “The modification… “ - I thought this is because HPV vaccines dont treat existing infections and that other medical interventions for the HPV related cancers have side effects.
Why are some words highlighted in green?
Page 3 “Moreover, Protein L2…” Which proteins are authors talking about? host or viral? Where does this cleavage happen? Either this sentence is very long for no reason or it doesnt make sense or both. This and the next paragraph need to be rewritten.
Figure 1 - I can barely read the labels in the pathway on the right. So what happens when the therapeutic vaccine injected? Does it result in cancer or does it halt cancer? Why do the precancer cells are shown to go back into the epithelial layers below? This figure has not been referred to in the actual manuscript below. Did the authors make this figure themselves or adapt it from an existing manuscript? If it is adapted, the original paper needs to be cited. If the authors made it using something like Biorender, then the authors need to cite that. Not citing anything is really bad practice.
Results line 3 - Figure 2 is flowchart. Figure 1 is completely different.
Discussion Line 4 - While there is nothing scientifically wrong with this statement, i would like to say that this view is now slightly outdated. Neutralizing antibodies are ONE of the markers for an immune response even for a vaccine. A good vaccine should stimulate both humoral and cellular immunity. More recent work on Ebola and Sars Cov2 has demonstrated that the non neutralizing antibodies actually have FC effector functions and are equally if not more important in providing protection against either an infection or an autoimmune disease like cancer. I would heavily encourage authors to seek out this literature about non neutralizing protective antibodies which in my opinion is a fascinating aspect of protection against any disease.
Page 12 “Vaccines against the …” - Is it vaccines against the L2 protein or vaccines that use the L2 protein as an immunological challenge?
References 40-42 - What was the out come of this test? Or did they demonstrate the successful tagging of the L2 protein to this nanoparticle?
Page 13 – Partial retrieval - How do you partially retrieve a protein from a biological sample? You can have no amount, low amount or high amount. You cannot have partial retrieval.
Page 13 – “Literature data supported…” – What is this different behaviour?
Page 13 “In particular…” Paragraph construction is very faulty. What is escape mechanism of viral persistence? I am not sure what the authors want to say about either cellular response or virus escape in HPV infected individuals.
Page 13 – “Finally… site of HPV infection” - Once again why is this information not in the introduction?
Page 14 “Therapeutic vaccines…” - incorrect clause. Authors suggest using the lesions in children or the vaccine to cure the lesions in children? Statement is very ambiguously worded. Also, the authors took out the point of treatment of HPV related cancers being very destructive and hence, a therapeutic vaccine may be a better option than chemotherapy of other cancer treatments.

Author Response

General comments

English continues to be a problem. I recommend an extensive round of proof reading. A new editing of english was perfomed by a native english speaker (USA) 1 Introduction – “For this motivation…” what do authors mean by motivation? Also, this highlighted portion still doesn’t tell me why therapeutic vaccines are the need of the hour. The author have corrected “for this motivation” with reason in order to explain the previous purpose. HPV causes different kind of cancer in different body areas. Therapeutic vaccines are designed to generate a specific anti-tumor response targeting cells expressing hpv antigens and decrease the risk of tumor development. 2 Page 3 “The modification… “ - I thought this is because HPV vaccines don't treat existing infections and that other medical interventions for the HPV related cancers have side effects. Why are some words highlighted in green? HPV vaccines aim to treat preexisting infections. Green words are only an error. 3 Page 3 “Moreover, Protein L2…” Which proteins are authors talking about? host or viral? Where does this cleavage happen? Either this sentence is very long for no reason or it doesnt make sense or both. This and the next paragraph need to be rewritten. The author have corrected the sentence. 4 Figure 1 - I can barely read the labels in the pathway on the right. So what happens when the therapeutic vaccine injected? Does it result in cancer or does it halt cancer? Why do the precancer cells are shown to go back into the epithelial layers below? This figure has not been referred to in the actual manuscript below. Did the authors make this figure themselves or adapt it from an existing manuscript? If it is adapted, the original paper needs to be cited. If the authors made it using something like Biorender, then the authors need to cite that. Not citing anything is really bad practice. We will increase the dimension of the carchters of the figure. The figure just shows when the vaccine starts to act in disease progression and not the outcomes. The Pathogenensis of cancer lesion start from the upper layer of the epithelium to the basal membrane. A new figure was prepared in order to clarify the virus pathway the authors cite Biorender. 5 Results line 3 - Figure 2 is flowchart. Figure 1 is completely different. The correction was made 6 Discussion Line 4 - While there is nothing scientifically wrong with this statement, i would like to say that this view is now slightly outdated. Neutralizing antibodies are ONE of the markers for an immune response even for a vaccine. A good vaccine should stimulate both humoral and cellular immunity. More recent work on Ebola and Sars Cov2 has demonstrated that the non neutralizing antibodies actually have FC effector functions and are equally if not more important in providing protection against either an infection or an autoimmune disease like cancer. I would heavily encourage authors to seek out this literature about non neutralizing protective antibodies which in my opinion is a fascinating aspect of protection against any disease. Several studies have shown the efficacy of not neutralizing antibodies but neutralizing antibodies seem to give stronger immunological response against different HPV genotypes and have a greater chance of preventing the spread of the lesion. Regarding neutralizing antibodies, the author appreciate the information provided by the reviewer, because these provides excellent food for thought also in case of HPV. 7 Page 12 “Vaccines against the …” - Is it vaccines against the L2 protein or vaccines that use the L2 protein as an immunological challenge? Vaccines use antigens of the pathogen to create an immunological response against HPV. 8 References 40-42 - What was the out come of this test? Or did they demonstrate the successful tagging of the L2 protein to this nanoparticle? In this study authors demonstrate that nanoparticles result as strong L2-based immunogens with a high potential as HPV vaccine, but also as a novel and extremely versatile peptide-antigen presentation platform. 9 Page 13 – Partial retrieval - How do you partially retrieve a protein from a biological sample? You can have no amount, low amount or high amount. You cannot have partial retrieval. The authors corrected the sentence

10 Page 13 – “Literature data supported…” – What is this different behaviour?

Behavior represents the different possible evolution of HPV lesion: evolution or regression.

11 Page 13 “In particular…” Paragraph construction is very faulty. What is escape mechanism of viral persistence? I am not sure what the authors want to say about either cellular response or virus escape in HPV infected individuals.

The authors corrected the sentence

12 Page 13 – “Finally… site of HPV infection” - Once again why is this information not in the introduction?

Because it permits to introduce what we explain after in the discussion

13 Page 14 “Therapeutic vaccines…” - incorrect clause. Authors suggest using the lesions in children or the vaccine to cure the lesions in children? Statement is very ambiguously worded. Also, the authors took out the point of treatment of HPV related cancers being very destructive and hence, a therapeutic vaccine may be a better option than chemotherapy of other cancer treatments.

The statement means that therapeutic vaccine should be used to promote the regression of laryngeal juvenile papillomatosis recurrent and to avoid several treatment

Round 3

Reviewer 1 Report

My compliments to the authors for making all the modifications. I would recommend one final round of proof reading and then the manuscript is ready for publication. This profreading can be done in house or by someone like the associate editor. No comments from my side necessary for this.

This manuscript is a resubmission of an earlier submission. The following is a list of the peer review reports and author responses from that submission.